# Aortic Valve Calcium Score: Applications in Clinical Practice and Scientific Research—A Narrative Review

**DOI:** 10.3390/jcm13144064

**Published:** 2024-07-11

**Authors:** Paweł Gać, Arkadiusz Jaworski, Filip Grajnert, Katarzyna Kicman, Agnieszka Trejtowicz-Sutor, Konrad Witkowski, Małgorzata Poręba, Rafał Poręba

**Affiliations:** 1Centre of Diagnostic Imaging, 4th Military Hospital, Rudolfa Weigla 5, 50-981 Wrocław, Poland; pawel.gac@umw.edu.pl (P.G.); aggnieszka52@gmail.com (A.T.-S.); konwit95@interia.pl (K.W.); 2Department of Population Health, Division of Environmental Health and Occupational Medicine, Wroclaw Medical University, J. Mikulicza-Radeckiego 7, 50-345 Wrocław, Poland; 3Healthcare Team “County Hospital” in Sochaczew, Batalionow Chlopskich 3/7, 96-500 Sochaczew, Poland; 44th Military Hospital, Rudolfa Weigla 5, 50-981 Wrocław, Poland; f.grajnert@icloud.com; 5Department of Paralympic Sports, Wroclaw University of Health and Sport Sciences, Witelona 25a, 51-617 Wrocław, Poland; 6Department of Internal and Occupational Diseases, Hypertension and Clinical Oncology, Wroclaw Medical University, Borowska 213, 50-556 Wrocław, Poland; rafal.poreba@umw.edu.pl

**Keywords:** CT-AVC, aortic valve calcification score, AVC, calcium scoring, aortic valve stenosis, bicuspid, multidetector computed tomography, echocardiography, artificial intelligence

## Abstract

In this narrative review, we investigate the essential role played by the computed tomography Aortic Valve Calcium Score (AVCS) in the cardiovascular diagnostic landscape, with a special focus on its implications for clinical practice and scientific research. Calcific aortic valve stenosis is the most prevalent type of aortic stenosis (AS) in industrialized countries, and due to the aging population, its prevalence is increasing. While transthoracic echocardiography (TTE) remains the gold standard, AVCS stands out as an essential complementary tool in evaluating patients with AS. The advantage of AVCS is its independence from flow; this allows for a more precise evaluation of patients with discordant findings in TTE. Further clinical applications of AVCS include in the assessment of patients before transcatheter aortic valve replacement (TAVR), as it helps in predicting outcomes and provides prognostic information post-TAVR. Additionally, we describe different AVCS thresholds regarding gender and the anatomical variations of the aortic valve. Finally, we discuss various scientific studies where AVCS was applied. As AVCS has some limitations, due to the pathophysiologies of AS extending beyond calcification and gender differences, scientists strive to validate contrast-enhanced AVCS. Furthermore, research on developing radiation-free methods of measuring calcium content is ongoing.

## 1. Introduction

The aortic valve, which prevents the return of blood from the aorta to the left ventricle, consists of three semilunar cusps: posterior, left, and right [1]. There have been documented cases with a unicuspid aortic valve [2], as well as ones consisting of five cusps [3]. The physiological function of this vital heart structure is both straightforward and complex. During the systolic phase, when the mitral valve is closed, the aortic valve cusps open, allowing blood to be ejected into the circulatory system. Conversely, during the diastolic phase when blood flows in the opposite direction—back to the left ventricle—the reverse blood flow presses the semilunar cusps together, preventing blood from returning [1]. Throughout the whole lifetime, the human body is exposed to many risk factors that could damage or weaken this fragile structure [4,5]. Aortic stenosis (AS) is a progressive disease that involves the narrowing of the aortic valve opening and it is the most common acquired valve disease in developed countries [6,7]. Previously, the majority of cases were caused by rheumatic fever [8]. Currently, in industrialized nations, the disease occurs most frequently via a degenerative process [9]. AS affects roughly 12.4% of elderly people [10], and the incidence rate of severe AS measures at approximately 4.4‰/year [11]. Cardiologists have noted that, with aging and in certain medical conditions, the cusps of the valve become calcified, contributing to the deterioration of its function [12]. Studies showed that, in 2019, around 9,404,077 people suffered from calcific aortic valve disease (CAVD) [13]. The population is aging; therefore, the prevalence of the CAVD and AS is still increasing [14,15]. The CAVD leads to calcific aortic valve stenosis (CAVS), which causes left ventricular hypertrophy, and finally heart failure [16]. AS is associated with major mortality and a high societal cost [17]. For that reason, early, effective, and affordable diagnostics are desirable. Initial symptoms, such as angina pectoris, dyspnea, decreased exercise tolerance, syncope, or a systolic ejection murmur during physical examination, may suggest the presence of AS [18]. A chest X-ray and electrocardiography can be helpful in the initial evaluation, as the former can show signs of heart failure and detect vascular calcification, while the latter may detect comorbidities [19]. Point-of-care ultrasound can visualize features of the disease, such as reduced leaflet motion or left ventricular hypertrophy [20]. Subsequently, the primary test for diagnosis and evaluation is transthoracic echocardiography (TTE), which allows for a comprehensive assessment of valve anatomy, structure, and hemodynamics [21]. However, varied clinical situations with inconclusive parameters and preinterventional planning require more thorough examination, indicating that TTE is insufficient in certain scenarios [22,23]. Different imaging methods provide additional diagnostic and prognostic information. Cardiac magnetic resonance (CMR) provides valuable information about the myocardium and myocardial fibrosis [24]. Cardiac computed tomography (CCT) clarifies disease severity in patients with discordant echocardiographic findings and provides information for surgeons before transcutaneous aortic valve implantation (TAVI). Therefore, a multimodality imaging approach is essential in the diagnosis and prognosis of CAVS [25]. Research on a scale that would allow for the estimation of the calcification grade of the aortic valve has been initiated. In these prosperous times of radiology and the fast development of medical equipment, we would like to be more precise and able to make therapeutic decisions early enough to help the patient. One of the tools used by cardiologists and radiologists is an aortic valve calcium score (AVCS), also called computed tomography aortic valve calcification scoring (CT-AVC). Is this scale ideal? How can it be used? What is its application in clinical practice and in scientific research? The aim of this review is to summarize the current knowledge of the application of AVCS, its effectiveness and limitations, and comparison with other methods.

## 2. Materials and Methods

For this review, focusing on the potential use of the aortic valve calcification score, we conducted a comprehensive literature search for articles published between 2018 and 2024. We searched the PubMed database using the following search query: (“AVCS*” OR “calcium scor*” OR “aortic valve calcification scor*” OR “Agatston score”) AND ((“clinical practice” OR “clinical trial*” OR “scientific research*” OR “Practice Patterns, Physicians”[Mesh]) OR (“aortic stenosis*” OR “aortic valve stenosis” OR “Aortic Valve Stenosis”[Mesh])). The specified query yielded 310 results. Additionally, we conducted searches in both PubMed and Embase using the following terms in various combinations: ‘AVCS’, ‘aortic valve calcification score’, ‘aortic stenosis’, ‘aortic valve stenosis’, ‘transcatheter aortic valve implantation’, ‘TAVI’, ‘transcatheter aortic valve replacement’, ‘TAVR’, ‘stroke’, ‘sex’, ‘hypertension’, ‘echocardiography’, ‘ultrasound’, ‘cholesterol’, ‘LDL’, ‘lipids’, ‘tobacco smoke’, ‘bicuspid aortic valve’. Furthermore, we retrieved relevant articles from the reference lists of included articles where applicable, regardless of the publication date. We included only articles written in English. The final number of papers selected for this manuscript was 207, with the consensus of the authors. With the aim of discussing the topic more holistically, we decided to forego the systematic review in favor of a narrative review.

## 3. What Is and How to Measure AVCS?

The degree of calcification is assessed using the Agatston method, originally developed in 1990 for coronary artery calcification assessment [26]. The concept of AVCS is similar to the primary method of measuring coronary calcium score. The amount of calcium in the aortic valve correlates with the severity of aortic stenosis [27]. Imaging data are acquired through non-contrast computed tomography scanning.

The procedure requires a specific protocol, including a voltage range of 120–140 kV, a tube current (mA) based on the patient’s body weight, a slice thickness of 2.5 to 3 mm, a matrix size of 512 × 512, and electrocardiogram-gated acquisition in diastole (60 to 80% of the RR interval). The radiation dose is low, measuring approximately 1 mSV. The threshold for calcium detection is 130 Hounsfield units (HU). To calculate the AVCS value, a weight is assigned depending on the level of HU attenuation of the considered lesion. A weight of 1 is assigned to an attenuation of 130 to 199 HU, a weight of 2 to 200 to 299 HU, a weight of 3 to 300 to 399 HU, and a weight of 4 to ≥400 HU. The weighting factor is multiplied by the area of the specific lesion in square millimeters, and values for all lesions are summed up. In this way, the total calcium score is obtained. The radiologist includes both valve leaflet and annular calcification. The measurement must be performed on axial slices, and multiplanar reconstruction views are valuable in the differentiation of valvular from non-valvular calcium and quantification of the calcification score using a leaflet [26,28,29,30].

However, there are potential pitfalls in AVCS assessment. Clinicians should be aware that some patients may have a continuity of calcification between the left ventricular outflow tract (LVOT) and the aortic valve, as well as between the mitral valve and the aortic valve. LVOT, proximal coronary arteries, and mitral valve calcification should not be included in the AVC score, as this would lead to an overestimation of disease severity in those cases. Therefore, exact quantification may be difficult. Recommendations exist, but there are no endorsed guidelines or software platforms for separating valvular calcium from non-valvular adjacent structures’ calcium [30]. Furthermore, studies indicate other aspects that clinicians may require information about. Hinzpeter et al. [31] investigated whether the usage of image reconstruction with iterative reconstruction (IR) techniques would affect AVCS evaluation and thus the classification of aortic stenosis severity by CT. IR techniques have the potential to enable imaging at lower radiation doses while preserving image quality. The conclusion is that increasing IR strength leads to a decrease in AVCS values, and thus incorrect categorization and underestimation of disease severity. Research has shown that aortic valve calcification measured using computer tomography is an accurate, repeatable, and well-validated indicator of the severity of stenosis [30,32,33]. The acquired scores highly correlate with the total calcium content in explanted valves [34,35]. Different platforms can generate slightly different absolute values; however, there is no statistical significance; therefore, AVC scoring may be considered software-independent [36], but more research is needed to ensure generalizability. In addition to assessing aortic valve calcification, computer tomography also provides us with other important information required for the procedure, such as the size of the aortic root, which is needed to appropriately select the prosthesis size, an evaluation of the access route morphology, and, if necessary, an assessment of coexisting diseases [37].

## 4. Mechanisms of Aortic Valve Calcification

Aortic valve calcification occurs when calcium deposits build up on the aortic valve, leading to the narrowing of the valve orifice, a condition known as calcific aortic stenosis (CAVS) [6,9,38]. The pathophysiology of the aortic valve can be divided into two separate phases—the initiation phase and the propagation phase. For many years, aortic stenosis was considered a passive, age-related, degenerative process based on calcium deposition. However, we now understand that it involves more complex biological and molecular reactions, inflammation, and fibrinolysis [39,40,41].

In the initiation phase, there is valvular endothelial injury and a process of inflammation, accompanied by lipid deposition and microcalcification. This active process resembles atherosclerosis and shares main risk factors such as high-LDL cholesterol, lipoprotein (a) [Lp(a)], smoking, and inflammation, triggering a cascade of reactions [42,43,44]. Hyperlipidemia, especially elevated Lp(a) levels, promotes local inflammation, and exposure to environmental cigarette smoke is directly proportional to increased calcification of the aortic valve [43]. Non-lipid mechanisms may also be important in the progression of aortic valve stenosis [45]; mitochondrial dysfunction, as an example, is implicated in oxidative stress, contributing to inflammation and releasing the primary molecular mediators of AS development, and also plays a role in regulating valvular remodeling [46]. The propagation phase involves a vicious cycle of calcification, starting with the valvular interstitium converting into an osteoblastic phenotype and initiating the calcification process. The amount of calcium increases over time, correlating with disease progression [6,39,40]. Regarding specific leaflets, in male patients, CTA-derived aortic valve leaflet calcification load (AVLC_CTA_) of the noncoronary cusp and right coronary cusp has a greater impact on AS severity than calcification of the left coronary cusp [47]. The calcification activity in the aortic valve, from mild to moderate stages, is primarily driven by the settled calcium burden [48]; because of that, the significance of other risk factors diminishes in more advanced stages of AS.

Lipoprotein (a) (Lp(a)) is a factor in cardiovascular disease that deserves special attention due to its pro-atherogenic and pro-inflammatory potential. It is strongly associated with the development of aortic stenosis [42,49,50,51,52,53,54,55,56,57,58,59,60,61] and the calcification process in the aortic valve [57]. Lipoprotein (a) consists of apolipoprotein (a) bound to apolipoprotein B-100. Lp(a) is a preferential carrier of oxidized phospholipids in its lipid phase [62], contributing to its pro-atherogenic and pro-inflammatory effects and the development of aortic stenosis [50,63,64,65]. However, the exact mechanisms by which Lp(a) causes aortic stenosis are not yet fully explained. As Lp(a) is proven to be a strong causal factor for AS, it potentially explains the residual cardiovascular disease (CVD) risk in patients with low LDL values and the optimal treatment of conventional risk factors [66,67,68]. Its higher association with the progression of CAVS than LDL [42,64,66] suggests that it could be a future target for treatment, preventing the progression of AS to severe aortic stenosis [69]. There is strong evidence that Lp(a) levels are genetically determined, making it a genetic risk factor for cardiovascular disease [62].

AS development is known to be associated with various conditions, including hypertension, metabolic syndrome, diabetes mellitus, and familial hypercholesterolemia, all of which are also causes of atherosclerosis [70,71]. Additional factors include aging, renal insufficiency, genetic mutations, and chronic inflammation [9,41,72,73,74,75]. Notably, familial hypercholesterolemia strongly correlates with the development of CAVS [76]. This condition is characterized by elevated levels of LDL, specifically Lp(a), suggesting an important causal association [66,67,77]. Calcification of the aortic valve is more common in hemodialysis patients, with a causal association observed between inflammatory markers such as CRP, fibrinogen, IL-6, and TNF-α. These markers predispose to inflammation, oxidative stress, and calcification, ultimately leading to CAVS [74]. A well-known factor associated with AS development and disease progression is diabetes mellitus (DM). AS is more prevalent in patients with DM than in the general population, which can possibly be explained by the accumulation of advanced glycation end products (AGEs) in the AV, which are associated with oxidative stress, calcification, and inflammation, in a stenotic valve [41]. Another significant risk factor for AS is male sex, possibly due to androgens. This may also explain the observed differences in the calcification pathway between males and females [38,78].

In summary, the mechanism of AS is a complex process divided into an initiation phase and propagation phase, involving valvular endothelial injury, inflammation, lipid deposition, and calcification, driven by factors such as lipoprotein (a) and cholesterol, and promoted by various diseases, leading to the narrowing of the aortic valve orifice and disease progression.

## 5. Diagnosis of Aortic Valve Stenosis

The initial step is the assessment of patient symptoms using the NYHA classification. AS patients often present with symptoms such as chest pain, shortness of breath, syncope, or heart failure. NYHA classes range from I (no symptoms) to IV (symptoms at rest). According to the ESC guidelines, symptomatic severe aortic stenosis has a strong recommendation for aortic valve surgical intervention [79]. The gold standard for assessing the severity of AS is TTE [80,81]. Echocardiography serves as the standard diagnostic tool for aortic stenosis due to its accessibility and speed. It is the best tool for measuring the hemodynamic values in aortic stenosis. To assess severe AS, measurements are taken of the mean gradient (≥40 mmHg), aortic valve area (AVA < 1.0 cm^2^), and peak aortic jet velocity (Vmax ≥ 4.0 m/s). These parameters have prognostic value for patients with AS and are used to divide patients into groups based on hemodynamics [82].

Another important parameter of TTE in assessing aortic stenosis is the effective orifice area (EOA), calculated from the velocity–time integral (VTI) of the LVOT and the measurement of the left ventricular outflow tract (LVOT). In truly severe aortic stenosis, LVEF can be reduced due to the small effective orifice area of a stenotic, calcified valve. Furthermore, the indexed effective orifice area (EOAi), calculated from EOA/Body Surface Area (BSA), is utilized as one of the most important echocardiographic indexes. It is used to measure the hemodynamic performance of the transcatheter heart valve (THV) and to characterize prosthesis–patient mismatch (PPM) [83,84,85].

An additional key aspect of AS diagnosis with TTE is the morphological assessment of the aortic valve and the visualization of calcifications. Two-dimensional echocardiography also serves as a diagnostic option by visually assessing AVS morphology and grading disease severity. A visual score (VS) ranging from 0 to 11 was developed and described in a study by Nemchyna et al. [86]. The study evaluates the aortic valve using VS, assessing leaflet mobility, echogenicity, thickening, and lesion localization. The results demonstrated a strong correlation between VS and disease severity, measured by CT-AVCS and hemodynamic values. Thresholds for detecting AS were established, with AS confirmed at a VS grade of 6, exhibiting high specificity in women (96%) and men (94%). This demonstrates that the use of VS to assess aortic valve morphology can serve as an important supplementary method for AS diagnostics and the evaluation of disease severity.

Various diagnostic imaging modalities can be employed independently, or as additional, complementary tools, to visualize and assess the severity of CAVS:Transthoracic echocardiography (TTE): This method is considered the first-choice diagnostic tool [79,83]. However, it is not infallible. If the results of TTE are inconclusive, other methods can be employed.Computed Tomography Aortic Valve Calcium Scoring: This method is particularly useful in patients with discordant echocardiographic results [87,88].Contrast-Enhanced CT: This method shows promising results in imaging fibrotic aortic stenosis in women and could potentially be a future method for assessing fibrotic aortic stenosis severity. However, for routine clinical use, it requires more robust evidence and validation [29].Cardiac Magnetic Resonance Imaging (CMR): This is a radiation-free and non-invasive method that allows for a thorough structural and functional evaluation, with the option to quantify flow [89].18F-NaF-PET: This method measures the uptake of 18F–sodium fluoride in the AV on images obtained through PET scans. This method may be useful for assessing prognosis, displaying regions of higher marker uptake, and identifying places of future calcium deposit accumulation [90,91].Dobutamine-Stress Echocardiography (DSE): This is a variant of TTE where patients are given augmenting doses of dobutamine intravenously every 3–5 min from 5 to 20 μg/kg/min. This provocation test can be used to assess patients with severe AS and discordant echocardiographic values [92].

## 6. Classification of Aortic Valve Stenosis Severity with AVCS

The severity of AS can be classified based on measurements of calcification using CT-AVCS threshold values, indicating the probability of severe stenosis [79]:Unlikely: AVCS < 1600, or, for women, <800, is classified as unlikely.Likely: AVCS > 2000 for men, or >1200 for women (Figure 1A), is considered likely.Highly Likely: AVCS > 3000 for men (Figure 1B), or >1600 for women, is classified as highly likely.

## 7. Sex-Specific Aortic Valve Calcification Score

The Agatston score exhibits variations between genders, with studies consistently highlighting strong gender differences in the search for threshold values for severe AS. Notably, CT-AVCS in women tends to be considerably lower than in men [29,32]. Various studies have converged on similar CT-AVC thresholds for severe AS. Pawade et al. [32] concluded that optimal threshold scores for developing severe AS are lower for women (1377 AU) and higher for men (2062 AU). These values align closely with those established by Clavel et al. in a multicenter study [93]—1274 AU for women and 2065 AU for men. Importantly, these values are akin to the threshold values of AVCS outlined in the 2017 guidelines of the European Society of Cardiology (ESC) and the European Association for Cardio-Thoracic Surgery (EACTS), which are 1200 AU for women and 2000 AU for men [94]. Most studies on AVCS were conducted in Western countries, emphasizing the need for research on populations from diverse regions. A retrospective study indicated that the CT-AVCS thresholds from the 2017 ESC guidelines for severe AS are applicable to the Japanese population [87]. More recent research by Boulif et al. [95] and the ESC guidelines from 2021 [79] suggest an incremental tendency in AVCS cutoff values. The summarized thresholds of different authors are presented in Table 1.

The higher threshold values observed in men, even when adjusted for body size, suggest that less calcification is required for the development of severe CAVS in women. This discrepancy may stem from the fact that AS in women more frequently involves a process predominantly characterized by fibrosis of the valve, which is not adequately quantified through AVCS. In contrast, the primary lesion in AS in men is calcification, leading to higher AVCS values. Consequently, the need for different cutoff values based on gender arises, with a lower AVCS score in women indicating a more severe disease compared to the same score in men [96]. Supporting this notion, Tribouilloy et al. [97] observed lower survival rates in women diagnosed with severe AS compared to the expected general population survival rates. Their study on gender differences in outcomes among patients with severe AS revealed an earlier onset of symptoms in women compared to men. Although women were diagnosed at a higher age, they exhibited greater symptomatology and fewer comorbidities related to AVS than men. Given these differences, it may be essential to consider a distinct approach to the assessment of AS in women [98].

## 8. Bicuspid Aortic Valve

Bicuspid aortic valve (BAV) (Figure 2A) is a condition characterized by the presence of two cusps in the aortic valve. This anatomical anomaly can lead to narrowing and calcification in the process of AS. BAV itself, an anatomical anomaly causing valvular asymmetry and hemodynamic disturbance, predisposes individuals to AS. Aortic stenosis tends to manifest at an earlier age in patients with BAV due to stress-induced calcifications resulting from asymmetrical leaflet motion and altered blood flow. Additionally, patients with BAV are more prone to developing aortic root and ascending aorta dilatation. The presence of AS in individuals with BAV necessitates a different diagnostic approach and an anatomical assessment of the valve before considering surgery [99].

Veulemans et al. [100] highlight that the AVC thresholds in guidelines do not distinguish between BAV and the tricuspid aortic valve (TAV) (Figure 2B). Their study assessed Agatston score, focusing on sex-specific differences in severe BAV-AS and TAV-AS. Their findings demonstrate that sex-specific AVC scores in severe AS were approximately 1/3 higher in patients with BAV than in those with TAV for both sexes. They conclude that there is a need for the adjustment of AVC thresholds in guidelines for BAV cases. On the other hand, findings from a retrospective single-center study indicate that men with BAV with severe AS have a higher AVC score and AVC density than men with TAV and severe AS. However, the calcification scores for both women with BAV and women with TAV do not differ significantly [101].

A study conducted by Wanchaitanawong et al. [102] proved that AVCS is an accurate method to assess AS severity in patients with BAV, but was not as successful in assessing rheumatic aortic stenosis, indicating that AVCS may not perform equally well across all etiologies of AS. Furthermore, as CT is a precise method for measuring Aortic Valve Area (AVA) (Figure 3) through planimetry and allows for the measurement of AVCS, it is particularly useful for assessing AS severity in patients with BAV [103]. AVA refers to the effective orifice area of the AV and is a crucial parameter when evaluating the severity of AS. In a study by Jeongju Kim et al., AVCS demonstrated a superior performance compared to TTE in identifying BAV. The study aimed to compare the diagnostic ability of CT and TTE and found that CT was more accurate in diagnosing AS in BAV, exhibiting higher sensitivity and a negative predictive value. The study also observed an association between a higher degree of valvular calcification in BAV and a decrease in the sensitivity of TTE [104]. In a study by Bo Hwa Choi et al., higher AVCS values were observed in patients with BAV. The estimated threshold values for severe AS in patients with BAV in this study were 1423 AU in women and 2573 AU in men [105,106].

## 9. Application of the AVCS

In numerous scientific studies, researchers are exploring the possibilities of the application of the AVCS.

### 9.1. Why Might AVCS Serve as an Alternative to TTE in Certain Situations?

Based on different values obtained in TTE, different subtypes of AVS, each presenting distinct hemodynamics, can be identified:High-gradient severe AS (HG-AS);Classic low-flow low-gradient AS (LFLG-AS);Paradoxical low-flow low-gradient AS (pLFLG-AS);Moderate AS.

LF-LG refers to a subset of patients with severe AS that do not meet the standard TTE criteria, as the mean gradient (MG, ≥40 mmHg) and peak jet velocity (PJV, Vmax ≥ 4.0 m/s) are lower due to the heart’s insufficiency to eject such a flow through the AV [30,32,107,108]. Therefore, it is important to note that the diagnosis of LF-LG AS may necessitate specific imaging protocols [109]. The classification based on TTE shares a commonality in that it depends on blood flow. In some cases, these values, specifically the mean gradient, can be discordant in symptomatic patients and insufficient to diagnose severe AS [22,110]. In an article by Kupeerstein et al. [82], the authors emphasize the significance of the underestimation of the true mean gradient, which can be falsely measured because of technical difficulties. In the scenario of discrepant findings, the patient is recommended to undergo AVCS assessment [22,107]. This is a common problem, as discordant data are a concern for many patients; it is estimated that 40% of patients undergoing standard TTE have inconclusive results [30,111]. This clinical uncertainty regarding patients has to be assessed by an alternative method, namely AVCS. That is the reason why AVCS has shown increasing significance and has been adopted as a method of assessing severe AS [23,88].

Veulemans et al. [112] performed a retrospective study in a group of patients with AS where only AVCS turned out to sufficiently distinguish moderate from severe AS in male patients classified as having severe LGAS. Furthermore, AVCS values in patients with severe pLG and LGAS were comparable. As mentioned earlier, TTE is successful in assessing the majority of cases of AVS. However, patients often have other coexisting conditions in addition to aortic valve stenosis, which can interfere with the interpretation of the results. It seems that TTE is insufficient to confirm AVS in those situations [81]. This occurs in cases where AS coexists with atrial fibrillation (AF) because AF alters blood flow. As a result, the parameter calculated in TTE, such as the mean gradient (MG), may underestimate the severity of AS. However, AVCS is measured through CT and is a parameter that is independent of flow, which means that it is not affected by the change in MG [96]. It is known that AF, which causes a low-flow state through the AV, is frequent in patients with LGAS, posing a challenge for TTE evaluation. However, AVCS scores showed equal disease severity in patients with severe AS and AF-LGAS compared to SR-HGAS. On the other hand, patients with symptomatic AF-LGAS had higher AVCS scores and higher mortality, which may suggest that AVCS and patient symptoms may be better indicators when assessing patient outcomes and prognosis in patients with AF-LGAS [113]. It was also observed that patients with AF and AS were more symptomatic and had a worse quality of life compared to those without AF [114]. Adham K. Alkurashi and colleagues [115] analyzed the values of AVCS and their relationship with MG obtained by TTE during AF compared to sinus rhythm (SR) in patients with severe AS who were undergoing transcatheter aortic valve replacement treatment. These patients underwent echocardiographic and CT examinations before the procedure and were divided into groups based on the presence of AF or SR. The results showed discordance between AVC and the MG in patients with AF compared to SR, particularly in men, suggesting that AF, leading to reduced blood flow, may underestimate the severity of AS when calculated using the MG. This in turn has the potential to impact the diagnosis of severe AS and delay interventional treatment, resulting in the too-late diagnosis of severe AS [23,107,110,115]. Timely intervention is crucial, as the progression of AS leads to remodeling of the heart muscle in patients, and ultimately, disease progression may result in death [22,110,116]. In situations where the application of TTE is limited, the usefulness of AVCS becomes apparent [82,115]. AVCS can eliminate uncertainty in challenging cases with comorbidities and confirm severe AS. It is, therefore, a crucial and significant tool in assessing inconclusive cases of AS [27,80]. Values above the threshold indicate severe aortic stenosis and serve as a method of stratifying patients, negatively impacting survival rates. Different values in both genders have comparable and independent prognostic values in terms of outcomes [117]. CT-AVCS exhibits a strong correlation with the risk of death and AVR in patients, independent of TTE results and age [32,95,96].

Apart from AVCS, other important parameters can be derived that can be used as criteria to assess AVS severity. One of them is the AVC density measurement (AU/cm^2^). AVC density is a better score for assessing bicuspid aortic stenosis because it minimizes AV size disparities. It is known that bicuspid valves are usually larger than tricuspid valves, potentially resulting in a higher AVCS. Measuring total calcium load would lead to AN overestimation of disease severity; therefore, measuring AVC density could avoid misinterpretations. However, the adaptation of this method requires further investigation [95]. Another advantage of AVCS is that it is more precise in measuring small differences over time compared to TTE [118]. AVCS is a very valuable tool to assess AS severity, providing a useful approach to cases with inconclusive echocardiographic results, and can confirm severe aortic stenosis [81,82,117]. The disadvantage is that it is obtained by CT, which exposes the patient to radiation. Additionally, it is less available than TTE, and therefore will not replace standard TTE. However, it remains an important tool with increasing potential that can be adopted for different applications [102].

Differences between these two methods are systematized in Table 2.

In cases of LG-AS (with a less than severe gradient and narrowed aortic valve area (AVA)), DSE is crucial for determining whether the reduced blood flow through the aortic valve can be reversed by increasing the contractility of the heart muscle and, consequently, the ejection fraction (EF), leading to an increased gradient [108]. DSE also allows for the measurement of contractility reserve (CR) in patients. Its presence has been proven to correlate with an increase in stroke volume index (SVI) and AVA after dobutamine provocation [119].

Severe AS in symptomatic patients can be further divided into stages (D1–D3) according to the American College of Cardiology/American Heart Association (ACC/AHA) guidelines [120] based on hemodynamic categories and the presence of symptoms [121]:Stage D1—severe AS (high-gradient AS, with MG > 40 mmHg independently of flow value and ejection fraction);Stage D2—severe AS (reduced EF, low-flow, low-gradient AS, also called classical LF-LG);Stage D3—severe AS (preserved EF, low stroke volume index (SVI), low-flow, low-gradient AS, also known as paradoxical LF-LG AS).

Evaluating patients in stage D1 is typically straightforward, as is treatment decision-making. However, DSE is regarded as inconclusive when there is a reduced LV flow reserve or no contractile reserve [122]. It was proven to be hard to confirm true disease severity for patients at stages D2 and D3. Stages D2 and D3 are more complicated to assess because of the discordant values between AVA and MG, and the use of DSE and TTE has limitations in those cases [108]. An algorithm was proposed by Cutting et al. [108] to assess LF-LG AS: first, a true low-flow state must be confirmed by measuring SVI through echocardiography. Secondly, the likelihood of severe AS should be assessed by AVCS according to up-to-date ESC guidelines. If the likelihood of severe AS is intermediate, and EF is reduced, DSE is recommended. On the other hand, if EF is preserved, nitroprusside testing should be performed. DSE is an alternative option to confirm AS severity in patients with a discordant TTE grading, according to Guzzetti et al., if the first-line method, namely AVCS, for verifying true anatomical severity is unavailable [92]. A study on 243 patients evaluated the usefulness of DSE for LG-AS assessment. DSE is presented as a valuable method for evaluating patients with LG-AS. The results demonstrated an increase in AVA in patients during DSE, regardless of whether they had high or low AVCS, suggesting that the aortic valve can retain its elasticity even in cases of severe AS. However, a higher AVCS was associated with lower AVA values before dobutamine administration. No association between survival rate and the loss of valve pliability was observed, thus limiting DSE’s prognostic value [119]. Kragholm et al. suggest that it would be valuable to combine CT and DSE information, as patients with concordant DSE and AVCS findings showed the best clinical improvement [122].

### 9.2. The Potential of Utilizing Contrast-Enhanced CT in Quantifying AVCS

The routine measurement of AVCS with ECG-gated CT angiography (CTA) would be highly desirable, as it would reduce the radiation dose and examination time. However, the accurate calculation of the calcium score is significantly hindered by the presence of a contrast agent, with attenuation exceeding 130 HU. Consequently, the contrast agent may mask certain calcified changes [30,123]. Therefore, the method still has constraints and challenges. A retrospective study on LG-AS and HG-AS patients demonstrated that AVCS measured in contrast-enhanced CT (ceCT) is significantly lower than the score obtained in non-contrast (nc) CT; therefore, it underperforms in the categorization of patients into the appropriate AS severity group, especially those with LG-AS [124]. Similarly, Kim et al. [123] showed that measuring calcium content in aortic valves, as one component of a device landing zone calcium volume (DLZ-CV), obtained via ceCT in patients undergoing transcatheter aortic valve implantation (TAVI), results in an incorrect estimation of calcium volume. They suggest that a scan-specific individual HU threshold provides the best approximation. A study by Gać et al. [125] indicates that the conclusive estimation of AVCS solely using of the angiographic phase of MSCT is possible, and would additionally expose patients to a lower dose of radiation. The researchers developed mathematical formulations which allow for the transcription of the CTA AVCS to the AVCS native, the score based on the native phase of CT. However, the authors emphasized that they used fixed thresholds, which can omit calcifications with a density below the threshold. Angelillis et al. proposed that, by measuring the HU in the LVOT using the automatic 3Mensio software, the enhancement in the contrast of the aortic valve can be estimated and, therefore, the thresholds for correct calcium identification can be fixed [126]. Furthermore, CTA may have another advantage. When using a novel method that quantifies fibrocalcific volume, contrast-enhanced CT can be especially favorable to non-contrast CT in a subgroup of AS patients with a significant contribution of fibrosis, particularly females [127]. Still, the calcium score obtained in ceCT is dependent on an intricate interplay between factors associated with contrast material, scanner parameters, and patients’ body size or cardiac output. As mentioned in other reviews [29,30,128], AVCS measured with ceCT scans cannot be clinically utilized to define AS severity until the validation of the method is completed.

Nevertheless, some researchers have already provided data for the validation of CTA-derived AVCS. Pandey et al. [129] proposed a ‘dynamic thresholds’ approach and provided a formula for predicting calcium content in aortic valves using contrast-enhanced CT. The method showed high accuracy and a strong correlation with non-contrast CT AVCS. Abdelkhalek et al. [130] developed a method based on the false positive rate (FPR) that offers an opportunity to overcome the variability of the effect of contrast material on HU attenuation. Additionally, they introduced a scheme for assessing calcific geometry and structure using topographic maps. Their FPR method demonstrated a highly significant correlation with AVC scores measured with non-contrast CT. Furthermore, their scheme could be a clinically significant tool for the precise assessment of regional calcific progression, thereby facilitating the monitoring of AS development and predicting perioperative complications. Finally, contrast CT can be used in clinical patient assessments before transcatheter aortic valve replacement (Figure 4). Flores-Umanzor et al. summarized, in a systematic review [128], different methods of calcium score quantification on ceCT in patients undergoing TAVI. The authors emphasize that, as the Agatston method omits certain areas of nonvalvular calcification, other ceCT scores may provide additional data for a more accurate evaluation of AS and TAVI outcomes.

### 9.3. Calcium Score, TAVR, and the Risk of Stroke

The methods for treating severe aortic valve stenosis include surgical aortic valve replacement (SAVR) and transcatheter aortic valve replacement (TAVR), most commonly performed via the femoral artery or left subclavian artery [131]. TAVR, also referred to as TAVI, is considered a less invasive approach [132]. It is an alternative to SAVR, particularly for patients who are considered too high-risk for open-heart surgery. With the increasing life expectancy of patients and the associated comorbidities, the importance of TAVI for treating aortic stenosis will continue to grow. There has been a substantial global increase in TAVI procedures, accompanied by a correlated rise in balloon aortic valvuloplasty procedures. This trend is evident both among individuals deemed unsuitable candidates for TAVI due to futility and in cases where balloon aortic valvuloplasty serves as an interim measure preceding a definitive treatment, such as TAVI [132,133]. Like any medical procedure, it is associated with complications. We are most concerned about adverse cardiovascular events and conduction disorders. One of the more serious complications after TAVR is stroke. Any catheter manipulation of the aortic valve may lead to periprocedural ischemic stroke or TIA [134]. Calcified or atherosclerotic debris may be dislodged and cause embolization. The stroke specifically associated with TAVI is termed acute ischemic stroke complicating TAVI (AISCT) [135]. It may affect approximately 2.4 % of patients during the first month after the procedure [136]. Perioperative post-TAVI strokes lead to a more than six times greater risk of 30-day stroke related mortality [137]. The biggest risk of AISCT is within the first 24–48 h due to the procedure. Of note, the utilization of cerebral protection devices during the procedure reduces the frequency of ischemic cerebral lesions [138,139]. Diagnosis is made by examining the patient and confirmed either by CT or MRI, although MRI is much more sensitive [134]. However, in the case of a calcified embolus, brain CT is a more appropriate modality [140]. Therefore, CT may be given priority in the initial evaluation of suspected patients. The treatment of AISCT focuses on thrombolysis and the later use of anticoagulants and antiplatelet drugs. Neurointervention, such as mechanical thrombectomy and intra-arterial thrombolysis, tends to have better outcomes than conservative therapy in patients with moderate or severe AISCT [135]. Intravenous recombinant tissue-type plasminogen activator (IV rt-PA) involves a higher risk of bleeding after the operation; however, there are reports of the safe and effective administration of rt-PA during the short-term interval after TAVI [141].

Michael Foley and colleagues [142] conducted a study in which they demonstrated that a higher AVCS was associated with a higher risk of stroke following TAVR procedures. The authors’ considerations led to the possibility of individually assessing patients before treating aortic valve stenosis to determine the choice of procedure and the potential inclusion of a CEPD (cerebral embolic protection device) during the procedure. They aimed to identify patients for whom such an approach would be beneficial. Knowing which patients are at a higher risk of stroke following TAVR, one can attempt to find methods to prevent strokes and appropriately qualify patients for the procedure, taking into account their elevated risk. On the other hand, in the non-TAVI population, there was no association with the risk of stroke [142]. A number of authors suggest that there is no connection between calcification and periprocedural stroke [143,144], while others reveal correlations between solid emboli during valve positioning and AVC, particularly in LVOT, with implications for procedural complications [145,146]. The uncertain correlation between AVCS and the associated risk of stroke prompts an exploration of divergent outcomes across cited articles. Ongoing investigations are committed to refining stroke risk mitigation strategies in TAVI patients characterized by elevated AVCS. Concurrently, comprehensive findings elucidate the intricate relationship between AVC and procedural outcomes in TAVR. The data advocate for a nuanced integration of calcification considerations in risk stratification and procedural planning. The imperative for further research with expanded cohorts is underscored to deepen the comprehension of these complex associations within the cardiovascular interventions domain. Moreover, Bayar et al.s’ research [147] introduces ascending aorta intima-media thickness as a potentially valuable and accessible technique for identifying patients at risk of embolic complications during TAVI. This work underscores the significance of considering alternative vascular markers such as aortic intima-media thickness, in conjunction with traditional measures like AVC, in assessing the risk of embolic events during TAVI procedures. In summary, these insights underscore the ongoing quest to elucidate the intricate relationships between AVCS, stroke risk, and procedural outcomes in TAVI, emphasizing the need for diversified risk assessment tools and expanded research endeavors in the evolving landscape of cardiovascular interventions.

### 9.4. Association between Calcification, TAVR, and Mortality

Adverse cardiovascular events and conduction disorders can contribute to the increased mortality following TAVR procedures. Can long-term mortality following TAVR be predicted using the AVCS indicator? In scientific research conducted by various authors, there is significant interest in predicting adverse events following TAVR based on AVCS results. However, the results of these analyses are not always consistent or unequivocal. In one study, the authors conducted such an analysis, also examining whether there are differences between genders in the impact of the degree of aortic valve calcification on long-term mortality following TAVR. The conclusions from the study were as follows: the impact of AVCS on mortality is gender-modified, with increasing AVCS values being more strongly associated with a higher mortality risk among women after TAVR compared to men. The authors of the study emphasized that the conclusion drawn from this research could initiate a discussion and lead to further engagement in considerations regarding the need for developing a comprehensive tool for assessing various mortality prediction indicators, as AVCS could provide one such tool. This has benefits in terms of TAVR procedure planning, risk stratification, and individualized patient care [148]. As women have a smaller calcification component in AS compared to men, and therefore have lower AVCS values at comparable severity, a study by Choi et al. arrived at corresponding findings. They demonstrated that patients with low AVC scores exhibited unique clinical features and a higher risk of long-term mortality following TAVR [149]. In a study conducted by Spaziano et al. [150], the outcomes revealed a noteworthy association between left ventricular outflow tract calcification and an elevated risk of one-year mortality or stroke following TAVR. It is crucial to note that this study specifically focused on female subjects, introducing a gender-specific dimension to the observed relationship. There is also a correlation between a quantified indicator of aortomitral continuity calcification (AMCC) and one-year mortality post-TAVR. The AMCC score was calculated by assessing calcification within the aortomitral continuity, including the aortic valve, left ventricular outflow tract, and mitral annulus/valve [151]. AMCC may serve as a valuable adjunct to prevailing risk assessment tools, emphasizing the importance of incorporating novel calcification markers into the predictive framework to obtain a more accurate prognostication of TAVR outcomes. However, in another study, the results indicate that the degree of calcification did not emerge as a prognostic indicator for mortality during the postoperative monitoring period [35]. Further investigation is warranted to elucidate the multifaceted factors contributing to post-TAVR mortality, with a particular focus on exploring the potential predictive utility of the level of HDL-C, the CRP, and the monocyte-to-HDL cholesterol ratio [152,153].

### 9.5. How Can Evaluating the Calcium Content Be Useful in Predicting Outcomes and Complications after TAVR?

In the study by Alexandre Gamet and colleagues [154], the researchers analyzed the impact of AVCS on predicting complications following TAVR. However, they chose to include only the newer generation of valves in their analysis. The valve replacement was performed on symptomatic patients with severe stenosis. Two models of the newer generation of valves were used: Edwards Sapien 3 and Medtronic CoreValve Evolut R. The results were analyzed focusing on two groups of complications: the first group of complications included the possibility of all-cause mortality or cardiovascular mortality one month after the procedure, and the second group of complications included conduction disorders such as the occurrence of high-degree atrioventricular block or new left bundle branch block, and the need for permanent or transient cardiac pacing one month after the procedure. The obtained results contradict the correlation between the aortic valve calcification score and both the first and second groups of complications. No prognostic impact regarding the preoperative assessment of aortic valve calcification on the occurrence of death, severe cardiovascular complications, or conduction disorders after TAVR using Sapien 3 and CoreValve was demonstrated [154]. It is possible that the key factor in this case was the use of valves from the new generation, as the results of this study were not isolated findings. Similar conclusions were reached by Akodad M and colleagues [155]—when using the older generation of valves, AVCS was an independent predictor of outcomes after TAVI, whereas no such relationship was found when using the newer generation of valves. This may indicate a decrease in complications as a result of the advancement and improvement of medical technology [155].

Another application of the above indicator may be in predicting paravalvular leak (PVL) as a complication after TAVR procedures. PVL results in residual aortic valve regurgitation. Depending on the size of the leak, this can have various clinical implications, including the development of heart failure [156]. The diagnosis of PVL is key to predicting the effectiveness of the TAVI procedure and its long-term outcomes. A standard imaging method is TTE, which allows for the assessment of heart function and the detection of PVL. A more detailed method is transesophageal echocardiography (TEE), used when significant PVL is suspected. Both multidetector CT (MDCT) and contrast-enhanced CT are useful for diagnosis and are the best techniques to study thoracic aorta anatomy. To accurately measure the volume of the leak and its impact on heart function, cardiac MRI should be used. There is also the possibility of using cardiac catheterization, such as angiography or pressure measurement, although these methods are less frequently used. The choice of diagnostic method depends on the individual patient’s needs and the availability of technology. Experts agree that the best method of diagnosis is multimodality imaging [157]. In the analyses of studies by various authors, there was a consensus: a high AVCS is positively associated with PVL after TAVR [155,158]. Conversely, a low degree of AVC is a good predictor for an improvement in PVL after TAVR [159]. Building upon this consensus, a retrospective study was conducted, with a similar correlation being observed between early TAVI complications and AVCS from pre-TAVI CT. The results indicate a correlation between a higher AVCS and the increased prevalence of vascular leaks and other vascular complications [37]. Moreover, a systematic review confirms that a higher AVC measured using contrast-enhanced MDCT and the Agatson method, independently of the quantification method, is associated with an increased risk of adverse outcomes following TAVI. These adverse outcomes include the implantation of a permanent pacemaker, PVL, and aortic rupture. However, the authors emphasize the lack of and the need for standardization of the calcium burden evaluation for contrast-enhanced CT scans. Such guidelines would allow for the routine prediction of outcomes post-TAVI [128]. Similarly, the amount of calcium directly correlates with the development of PVL in the study by Kofler et al. [156], where they developed a reliable calcium score based on contrast-media-enhanced CT to predict PVL in patients undergoing TAVI. The score can be useful in improving risk stratification for this group of patients. Furthermore, other researchers in a retrospective study [160] present an ultra-low radiation dose and contrast volume protocol with dual-source CT. Their results indicate that the obtained AVCS is an independent risk factor related to major cardiovascular events after TAVI. In summary, a higher AVCS is consistently linked to an increased risk of paravalvular leak (PVL) and other adverse outcomes following TAVI procedures. This highlights the importance of standardized calcium burden evaluation methods to enhance risk stratification and outcome prediction.

Ciara Mahon et al.’s [161] investigation indicated that the individual valve leaflet calcification score may serve as an independent risk factor for paravalvular regurgitation, which also can be valuable for risk stratification. It is worth mentioning that the impact of the TAVR procedure, particularly the choice of balloon, on PVL is underscored by relevant studies. The use of the fourth-generation balloon-expandable transcatheter valve (4G-BEV) is linked to a decreased incidence of paravalvular leak at the 30-day post-TAVR mark when compared to the third-generation balloon-expandable transcatheter valve (3G-BEV) [162]. Some patients with more than mild PVL require a post-dilation (PD) procedure after TAVI. Uebelacker et al. aimed to determine the value of the AVCS and the peri-interventional aortic regurgitation (ARI) ratio in predicting PVL requiring PD. Their results indicate that a higher AVC score before TAVI predicts the demand for PD. However, they conclude that the ARI ratio would be more valuable as a predictive tool in this case [163]. A pre-TAVR assessment can be of significant value. When it comes to TTE, a retrospective analysis of pre-intervention measures revealed that the mean transaortic gradient serves as a predictor for post-TAVR quality-of-life enhancements. The study advocates for the use of MDCT in grading aortic stenosis severity in addition to TTE [164]. Another study suggests a significant correlation between the iodine concentration within the aortic valve complex, as assessed via spectral CT, and the extent of AVC volume reduction subsequent to TAVR signifies that quantifying the iodine concentration provides a valuable means for evaluating the propensity for AVC deformity following TAVR. For that reason, further clinical investigation is needed to determine if spectral CT should be performed before TAVR [165]. Several studies suggest that the pre-procedural assessment of aortic calcification helps guide the TAVI procedure, thereby improving the outcomes. AVCS is associated with an increased risk of post-dilatation after TAVI and atrioventricular block post-TAVI [166,167,168]. Moreover, the severity of AS, whether very severe (VSAS) or severe (SAS), does not significantly impact short-term mortality, but VSAS patients may require more interventions during TAVI.

This suggests that the severity of AS may not be the sole determinant in the decision-making process for TAVI [169].

By utilizing the AVCS indicator before TAVR procedures, clinicians can obtain valuable clinical information for surgical planning and anticipate potential adverse events. Based on this information, one can better plan the procedure itself and select patients in a way that maximizes the benefits of the treatment. This also creates opportunities for further improvements in the procedure. The application of the indicator discussed in this work is not limited solely to aspects related to the TAVR procedure. Furthermore, while AVCS remains a prognostic factor for aortic regurgitation with new-generation self-expandable valves, it does not singularly predict severe adverse outcomes. However, this conclusion is drawn within the context of various limitations, such as a relatively small sample size and variations in patient risk profiles [155]. It is imperative to acknowledge that AVCS is not the exclusive predictor, as recent research suggests the potential utility of insulin-like growth factor binding protein 2 (IGFBP-2) as a biomarker for predicting outcomes in TAVI patients, possibly surpassing existing scoring systems [170]. This highlights the evolving landscape of predictive markers in the field of TAVR, offering avenues for further refinements in patient management and outcome prediction. With the aging population, the prevalence of patients undergoing TAVR is on the rise, leading to a growing number of cases that require assessment [96,171]. Nevertheless, as with any surgical procedure, achieving and maintaining optimal clinical outcomes with TAVI requires significant operator experience [172]. It is important to note that the efficacy and complications of TAVI are influenced not only by aortic valve calcification but also by valve morphology and the patient’s comorbidities.

## 10. Application of AVCS in Research

In numerous scientific studies, researchers are exploring the possibilities of the application of AVCS. It has found use in research as a tool for comparisons of patient groups in multiple studies on diverse ethnic and gender populations. Furthermore, it has been adapted to evaluate patient outcomes and investigate risk factors for AVC.

### 10.1. Research on the Association between Calcium Content and Race, Ethnicity, and Gender

Several studies provide interesting findings regarding the association between race, ethnicity, and calcium content measured by CT-AVC. A study in a racially mixed sample suggests that racial background does not have a significant impact on the AVCS clinical thresholds. Men with severe AS had higher AVC scores than women with severe AS, irrespective of racial background [173]. Ellen Boakye et al. [174] used AVCS to compare the differences between median AVCS values in black males and females and white males and females. The white male group had the highest prevalence of aortic stenosis (58.2%), which, as expected, correlated with the highest median AVCS value (100.9 AU). Black males were the group with the second highest aortic stenosis prevalence (40.5%) and also had the second-highest median AVCS value (68.5 AU). Females had a lower prevalence of aortic stenosis and AVCS than males, and the black female group had the lowest prevalence of aortic stenosis (36.8%) and median AVCS (46.5 AU). Data acquired with the Agatston method can be further employed in studies investigating gender in terms of AS pathophysiology. Jover et al. [175] retrieved data from CT scans conducted on patients prior to the surgical valve replacement. The calcium score data revealed a correlation with neutrophil gelatinase-associated lipocalin (NGAL) expression in the AV in males and females. NGAL is associated with processes leading to AS, such as inflammation and calcification. The authors concluded that men showed a higher expression of NGAL, which may be responsible for the more significant progression of AVC in men.

### 10.2. Research on Improving Specific Diagnostic Methods

AVCS is utilized in research investigating the effectiveness of diagnostic methods in certain medical situations. Sancar et al. [176] used the calcium score to determine the severity of AS in patients with inconclusive DSE results. This classification helped to assess the diagnostic role of “Acceleration Time” (AT) in the estimation of true and pseudo-severe AS. AT showed a high predictive and discriminative strength in the estimation of true severe AS in patients with LF-LG AS. The aortic valve Agatston calcium score based on MDCT scans was used in parallel with DSE to determine the efficiency of catheter-induced premature ventricular contractions during postextrasystolic potentiation (PESP) mean gradient in detecting severe LG AS. The study showed that the method has a 100% sensitivity [177]. AVCS was also utilized for the validation of the utility of different ultrasound-based visualization methods and novel software. A comparison of the measurements of CT-AVC with those provided by automated 3D transesophageal echocardiographic software (Aortic Valve Navigator [AVN]) indicated the feasibility and accuracy of the AVN [178]. Ash et al. used AVCS and sex-specific AVC thresholds to adjudicate severe or non-severe AS in a study assessing the clinical utility of CT-derived aortic valve areas (AVAs); however, their results point out that the method has a poor capacity for distinguishing AS severity [179]. The score that is the subject of this paper was likewise used for the comparison with AS grading performed by echocardiographic visual score (VS). This helped in determining the validity of the VS, which is a visual assessment of AV morphology using two-dimensional echocardiography. This method could serve as an additional diagnostic tool for the detection and estimation of AS [86]. Data obtained with CT-AVC may also be used in the research as an additional element in a complex comparison of patients’ outcomes after AVR. In one such study, patients with elevated exercise pulmonary capillary wedge pressure (PCWP) ≥ 28 mmHg 1 year after the AVR procedure had a higher AVCS, but this was not statistically significant [180].

### 10.3. Research on the Possibility of the Incidental Detection of AS through CT for Other Indications

In a retrospective study, Ouchi et al. [181] aimed to determine AVCS threshold scores that could help in the early detection of significant asymptomatic AS incidentally during non-ECG-gated CT scans performed for non-cardiac indications. The acquired AVCS scores in this study were 2130.0 AU for severe AS and 1079.78 AU for moderate AS in patients with concordant TTE results. In patients with discordant TTE results, thresholds were estimated to be 1266.2 for severe AS and 474.5 AU for moderate AS. Those values displayed high specificity and sensitivity for detecting severe AS. Moreover, researchers explored the utility of evaluating AVCS in people undergoing a low-dose CT (LDCT). It was found that the LDCT screening program in lung cancer screening provides an economical opportunity for the early detection of AS using AVCS. Data synthesized with the method emerge as a substantial prognostic indicator for AS severity and cardiovascular disease (CVD) mortality [182,183,184,185]. On the other hand, in a retrospective analysis of patients by Brodov et al. [186], where the utility of entire thoracic arterial calcifications (TAC), CAC, and AVCS in predicting all-cause mortality (ACM) was evaluated, both CAC and AVCS were not found to be predictors of ACM. However, the TAC score emerged as a strong predictor of ACM.

### 10.4. Research on Environmental Risk Factors

The AVC scale also served to demonstrate the link between exposure to environmental tobacco smoke (ETS), estimated on the basis of the Second-Hand Smoke Exposure Scale (SHSES) questionnaire, and calcification in individuals with arterial hypertension (AH). To qualify for the study, patients had to have hypertension for at least 18 years and to have been pharmacologically treated for a minimum of 5 years. After analyzing the trial, the authors concluded that calcification of the aortic valve is not a passive process but a result of many biological and molecular reactions, inflammation, and fibrinolysis. Hyperlipidemia, specifically elevated Lp(a) levels, also plays a role in the calcification process by promoting local inflammation. Exposure to environmental cigarette smoke is directly proportional to an increase in the calcification of the AV [43].

### 10.5. Research on the Utility of the Score for Predicting Adverse Medical Events

Patients with presumed coronary artery disease underwent cardiac CT, and the acquired calcium score data were analyzed in a prospective cohort study. Researchers examined whether AVCS could serve as a predictor of major adverse cardiovascular events (MACE). The score showed the potential for predicting MACE incidence and was mostly associated with specific cardiovascular risk factors, such as hypertension, dyslipidemia, and age. However, the coronary artery calcification (CAC) Agatston score showed the strongest association and the greatest potential as a predictive factor [187].

### 10.6. Research on Concomitant Diseases and Risk States

The CT-AVC score was used in many studies to determine the association between AVC, AS, and simultaneous medical conditions. A population-based cross-sectional study investigated the effect of ambulatory blood pressure (BP) on AVC and CAC. The results indicated the association of AVC with office systolic BP, awake diastolic BP variability, and asleep systolic BP variability [188]. In another study [4], the researchers assessed how resting heart rate (RHR) is connected to the calcification of aortic valves, and they found that a higher RHR is correlated with AVC progression; this association was stronger for men and older adults. However, they did not establish causality. Yokohama et al. retrospectively investigated the CT data of patients with severe AS and demonstrated that AVCS significantly increased as the severity of concomitant aortic regurgitation (AR) increased [189]. Going further, a prospective longitudinal study compared mitral annular calcification (MAC) with the CT-AVC of patients with calcific aortic valve disease. The results of the study indicated a significant association between MAC prevalence and AVCS [190]. On the other hand, a study focusing on epicardial fat tissue (EFT) did not find a correlation between EFT volume and the Agatston score of the AV [191]. Chevance et al. evaluated the impact of valvular calcifications in patients with infective endocarditis (IE) and reported that patients with aortic IE and a higher aortic CT calcium score more frequently presented with atypical bacteria, that is, non-staphylococcus and non-streptococcus bacteria, on blood cultures [192].

Concerning diabetes mellitus (DM), the Agatston method of calculating AVC was utilized to demonstrate the higher prevalence of AVC in patients with DM [193,194]. Mourino-Alvarez et al. summarized the evidence for a relationship between DM and AS [195]. Hyperglycemia, insulin resistance, and dyslipidemia play a significant role in the calcification of the aortic valves. Furthermore, a clinical case series study [196] reported important findings in three patients with transthyretin (TTR) cardiac amyloidosis (CA) who had concomitant severe AS. Those patients had AVCS values below the ESC threshold for severe AS. Additionally, those values were in the range in which AS would be considered unlikely. This indicates that more research on patients with CA is required to establish novel Agatston aortic valve thresholds for the classification of AS.

Last but not least, Hu et al. conducted a cross-sectional study on patients with systemic lupus erythematosus (SLE) and used the Agatston method to determine the relationship between SLE and aorta and coronary artery calcification. Their results indicated that aorta calcification in this group of patients tended to occur at a younger age and was positively correlated with age [197].

### 10.7. Research in Hematology

The utilization of AVCS has been demonstrated in the assessment of hematologic patients. Philadelphia-negative Myeloproliferative Neoplasms (MPN) manifest a salient and sustained correlation with AVC, a relationship that maintains statistical significance even following adjustments for cardiovascular risk factors. AVC emerges as a potentially pivotal prognostic indicator for both prospective cardiovascular disease (CVD) and overall mortality in individuals with MPN. The inquiry unveiled an increased incidence of cardiac calcification within the MPN cohort, impacting both the coronary arteries and the aortic valve [198].

### 10.8. Research on the Efficacy of Medications

The score, which is the topic of this paper, can be similarly employed in studies investigating the effectiveness of drugs. Pawade et al. conducted a single-center double-blind randomized controlled trial on patients with calcific AS, in which they assessed whether denosumab or alendronic acid can inhibit the pathways leading to valvular calcification and therefore reduce the progression of AS. One of the tools for the patients’ evaluation was CT-AVCS. They concluded that both of the investigated drugs did not affect the progression of AVC [199].

### 10.9. Research on Association between Lipids and AS

In the scientific research, AVCS has been used as a tool to assess whether cholesterol levels or LDLR mutation are a better predictor for the occurrence of severe AVC. Patients suffering from familial hypercholesterolemia with mutations in the LDLR gene are characterized by elevated levels of total cholesterol and low-density lipoprotein fraction (LDL). The link between aortic stenosis and hyperlipidemia is well known and documented. However, the study mentioned here diverges from the others as the authors compared two groups with high cholesterol levels that differed regarding the presence or absence of mutations in the low-density lipoprotein receptor gene. The study was based on a comparison of AVCS in patients with hypercholesterolemia and a genetically diagnosed familial hypercholesterolemia with a mutation in the low-density lipoprotein receptor gene (LDLR-M) and patients with hypercholesterolemia without the gene mutation (LDLR-WT). The results indicated that AVCS was significantly higher in the LDLR-M group compared to the LDLR-WT group. By comparing the results of the AVCS, the authors concluded that AVC cannot be fully explained solely by cholesterol levels. Furthermore, the authors considered whether the role of LDLR in vascular calcification could also be linked to mechanisms other than lipid levels. They concluded that calcification of the AV cannot be fully explained by cholesterol levels; non-lipid mechanisms may also be involved in the progression of aortic valve stenosis. In this study, the LDLR mutation was a strong predictor of a high AVCS, but not cholesterol levels. LDLR appears to be a better predictor of severe AVC compared to cholesterol level measurements before and during treatment [45].

The relationship between AVCS and lipoproteins is also a subject of research. A study conducted on a cohort of Japanese individuals demonstrated that nuclear magnetic resonance (NMR)-quantified lipoprotein particles may serve as indicative markers for atherosclerotic diseases, specifically AVC [200]. Another study highlighted the pivotal role of low-density lipoprotein cholesterol (LDL-C) in influencing the protective advantages conferred by PCSK9 variants against calcific aortic valve stenosis onset [201]. The effect of HDL values on the progression of AVC in a multi-ethnic study of atherosclerosis was studied by Anna E. Bortnick et al. [202]. In this research paper, AVCS was used to evaluate patient follow-ups. The examined population had up to three CT scans over a median of (8.9 years). Elevated concentrations of high-density lipoprotein cholesterol (HDL-C) and high-density lipoprotein particles (HDL-P), characterized by a larger particle size and the absence of apolipoprotein C3 (apoC3), are notably linked to a reduced occurrence and advancement of AVC. This association remains significant even after adjusting for established cardiovascular risk factors. These findings suggest a protective role for HDL, in terms of AS progression and incidence, potentially attributed to its involvement in reverse cholesterol transport, protein cargo, or other pleiotropic effects. Conversely, the observed positive correlation with apoC3-containing HDL-C may indicate an increased susceptibility to AVC. Duygu Kocyigit et al. used AVCS to assess HDL cholesterol efflux-promoting function, revealing an impaired HDL cholesterol efflux capacity in moderate–severe calcific aortic valve stenosis [203].

Elevated Lp(a) and oxidized phospholipid (OxPL) plasma levels are associated with increased valvular calcification activity and faster disease progression in AS patients. Hence, there is a demand for clinical trials that would assess medications that could reduce the elevated Lp(a) and OxPL levels, as they could slow disease progression and delay the need for TAVR [204]. Higher Lp(a) concentrations are robustly associated with increased AVC, suggesting that screening high-risk individuals for AVC with CT imaging could be a fruitful approach [205]. Examining the impact of Lp(a) levels on aortic valve calcification activity using positron emission tomography/CT with sodium fluoride (18F-NaF), no significant difference between low-Lp(a) and high-Lp(a) patients was observed in terms of 18F-NaF uptake by the valve. Simultaneously, the amount of calcium in the valve emerged as the primary determinant of 18F-NaF uptake [48]. On the other hand, a study in the Rotterdam cohort found that Lp(a) is associated with baseline and new-onset AVC, emphasizing the potential effectiveness of Lp(a)-lowering interventions in the pre-calcific stages of aortic valve disease [206]. Finally, there is also a potential correlation between adipokine levels (leptin, resistin, and adiponectin) and the occurrence and severity of AVC, highlighting the complex interplay involving adipose tissue, adipokines, and cardiovascular health in the context of AVC. Further research is needed to comprehensively elucidate the underlying mechanisms and implications for CVD [207].

## 11. Alternative and Future Diagnostic Indicators of Calcification

Researchers are exploring alternative or novel approaches to assess CAVS.

As CT-AVCS involves exposing the patient to radiation and TTE may yield discordant findings in some patients, an ultrasound-based calcium scoring method validated by Gillis et al. [208] may be a promising technique that combines the advantages of TTE and CT-AVCS. It analyzes transthoracic echocardiographic images to derive the Global Calcium (GC) score and Calcification Selection Area. The GC score significantly correlates with the Agatston score, and additionally, it might assess dense fibrosis, contributing to the severity of AS, whereas CT-AVC focuses solely on calcification. Nevertheless, trials involving large cohorts are necessary to establish this method’s clinical utility with stronger evidence. A single-center study examined the utility of an AVC-3DEcho score, offering a chance to assess AVC with bedside three-dimensional TTE. The score correlated with the calcium weight obtained from a pathologic analysis of explanted valves and CT-AVCS. This pilot study suggests that the method could serve as a fast, non-invasive, complementary, and supportive means of identifying patients with severe AS, particularly those with discordant echocardiographic parameters [209]. Hirschberg et al. evaluated an echocardiographic calcification score (echo-CCS) in patients with low or intermediate cardiovascular risk, providing another radiation-free method. Their results demonstrated a good correlation with the Agatston score, indicating that echo-CCS could have potential use in predicting cardiac events, coronary interventions, and hospitalizations [210]. Another method of evaluating AVC with echocardiography was proposed by Tang et al., who developed a dynamical local feature fusion net capable of processing echocardiography to recognize AVC automatically. As it showed promising results, it could be another solid alternative to CT scans [211].

Furthermore, Hokken et al. validated the 3mensio Structural Heart package and compared it with a conventional manual calcium quantification tool. This package is a novel, semi-automated calcium quantification scoring module, involving a combination of the ECG-gated ceCT scan with a non-contrast scan. Combining reconstructions from both contrast-enhanced and non-contrast scans has the potential to streamline the identification of anatomical boundaries, thereby enhancing the precision of calcium quantification scores. Their results indicate that the AVC obtained with this method highly correlates with a standard reference, i.e., a conventional manual calcium scoring tool measured with IntelliSpace software [212]. It is also possible to measure AVCS through virtual noncontrast (VNC) images from photon-counting detector CT (PCD-CT), which shows promising results in terms of accuracy. VNC allows for the reconstruction of images from contrast-enhanced scans, eliminating the need to expose the patient to an additional scan. The authors conclude that the method enables the reliable quantification of AVC and risk stratification of patients. Future research is required to confirm its prognostic value [213].

Another noteworthy method is 18FNaF-PET. In contrast to AVCS, which depicts current calcifications in the aortic valve, 18FNaF-PET is a diagnostic method that provides insight into the future. By measuring the uptake of 18F–sodium fluoride in the aortic valve on images obtained through PET scans, this method can display regions of higher marker uptake and places where calcium deposits may accumulate. These regions are not detectable through AVCS because there is no calcium plaque, but marks where calcification will occur in the future [90,91]. It is a promising diagnostic method that could help quantify microcalcifications in vivo in the aortic valve before the onset of disease. In this way, it could predict future disease development, particularly in younger patients with early-stage disease, and offer prognostic implications for patients at risk of developing calcific aortic valve stenosis [24,90]. On the other hand, one study suggests that 18F-NaF PET-CT is not a crucial diagnostic method to predict severe AS development, although it could perform well in predicting the hemodynamic progression of CAVD in patients with a tricuspid aortic valve [214].

## 12. Will the Future of AVCS Quantification Be Dominated by AI?

The rapid development of technology, specifically artificial intelligence (AI) involving machine learning algorithms (ML), may also become very useful in medicine. Pradyumna Agasthi et al. predict, based on their results regarding the usefulness of the gradient boosting machine learning (GBM) model in predicting one-year mortality post-TAVR, that if the development of technology in medicine continues to accelerate, machine learning (ML) will grow in significance in clinical use [215]. Jonathan J. Hyett Bray et al. also highlight the increased significance of ML, which has great potential when incorporated in calculation, for example, of the AVCS. It would be an exciting advancement if the calculation of CT-AVCS, which is necessary for making decisions and selecting patients for TAVR, becomes automated, thus saving both time and resources [216]. These algorithms are already being designed at present. In 2021, Suyon Chang et al. aimed to create an algorithm for deep learning (DL), a DL-based algorithm that automatically quantifies AVCS from CT images. The DL-based results were then compared to visually assessed AVCS by radiologists to evaluate the accuracy of this method. Interestingly, the AVCS measured by DL algorithms had a higher accuracy (92.9%) than the measurements made by radiologists (77.8–89.9%). The diagnostic performance of automatic DL measurements of AVCS can, therefore, outperform the manual assessment of AVC severity classification [217].

## 13. Summary

The success of imaging diagnostics lies in their precision and reproducibility, acknowledging the absence of an ideal diagnostic method due to the unique nature of each patient’s medical case. In cases where traditional assessment methods pose challenges, particularly in patients with concomitant diseases and discordant echocardiographic results, CT-AVCS emerges as a powerful diagnostic aid with several advantages, including its independence from blood flow and more precise assessment of valve morphology compared to TTE. It exhibits high reproducibility and sensitivity, enabling the detection of subtle changes over time and facilitating comparisons of the results. Its predictive value extends to outcomes related to aortic valve stenosis progression, mortality, patient outcomes, and the necessity for valve replacement. Moreover, AVCS proves particularly useful in inconclusive scenarios, such as in low-flow, low-gradient, and paradoxically low-flow low-gradient patients, which constitute a significant proportion of the cases with AS, accounting for 40% of instances. Factors such as diet, environmental exposure, and genetics impact Calcific Aortic Stenosis either positively or negatively. The quest to predict calcification based on environmental tobacco smoke exposure or evaluate the risk concerning diet and genetic mutations highlights ongoing challenges. Although research has been conducted since the 1990s, instances remain where the calcification results do not statistically predict adverse outcomes, emphasizing the imperfections in the current scale. Equipment quality and the medical doctors’ skill set further influence the results, raising questions about the human ability to match technological advances in radiology. While it is improbable that AVCS will replace TTE in the future, its advantages position it as a valuable complementary method, especially in cases of severe aortic valve stenosis where a comprehensive assessment is crucial. The future of assessing patients with aortic stenosis is envisioned, involving the adoption of a multimodality approach and integration of various diagnostic methods. The goal of this holistic approach is to facilitate individualized patient approaches and expedite diagnostics to obtain positive outcomes.

## Figures and Tables

**Figure 1 jcm-13-04064-f001:**
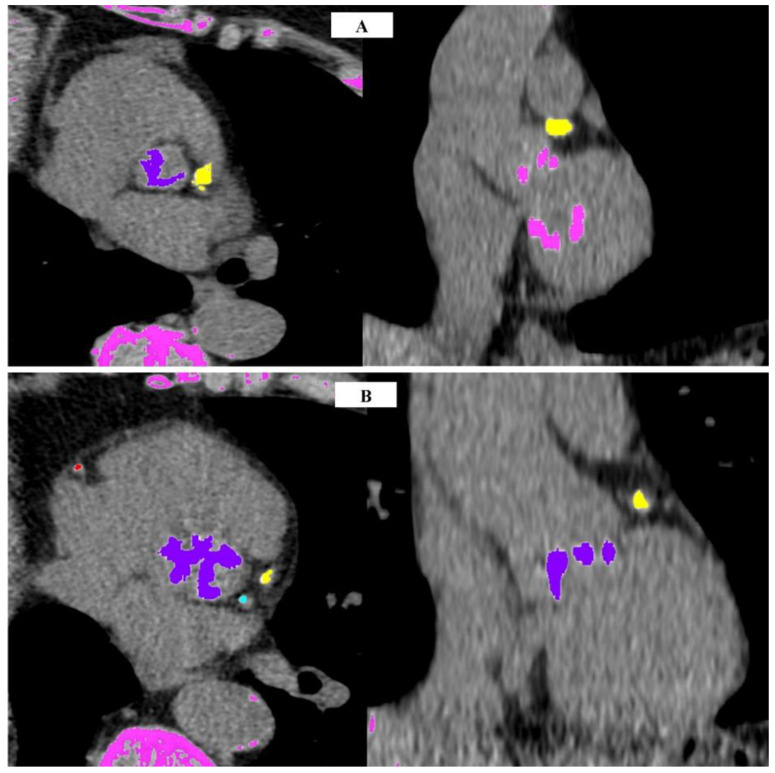
Examples of cardiac computed tomography images in the AVCS assessment protocol: (**A**) 67-year-old woman with AVCS 1578; (**B**) 74-year-old man with AVCS 5323 (images from P.G.’s clinical practice). Purple—calcifications in the aortic valve, yellow—calcifications in the LAD branch, blue—calcifications in the LCx branch, pink—calcifications in other anatomical structures, red—calcifications in the RCA.

**Figure 2 jcm-13-04064-f002:**
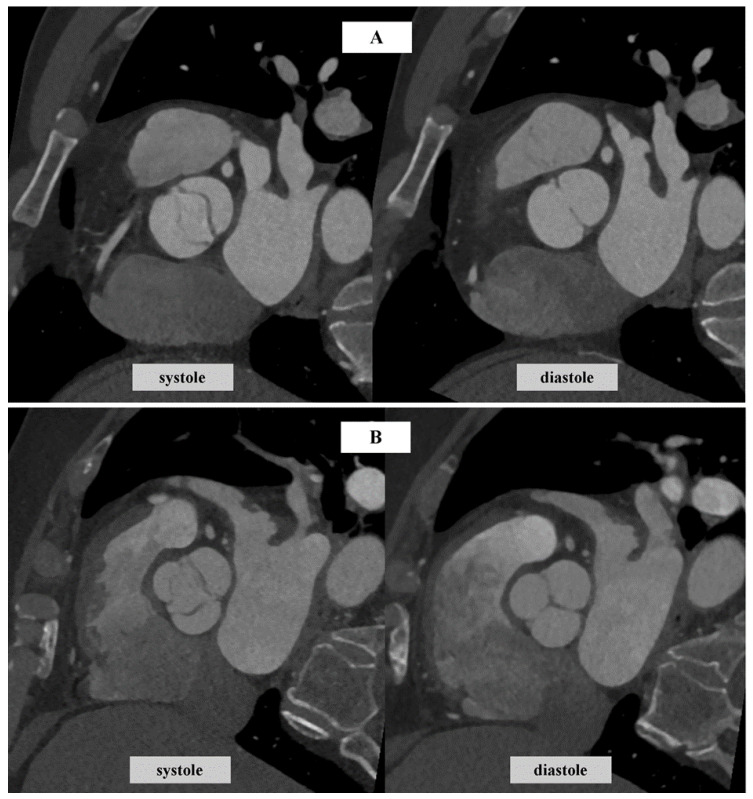
Sample images of the aortic valve in cardiac computed tomography: (**A**) bicuspid aortic valve; (**B**) tricuspid aortic valve (images from P.G.’s clinical practice).

**Figure 3 jcm-13-04064-f003:**
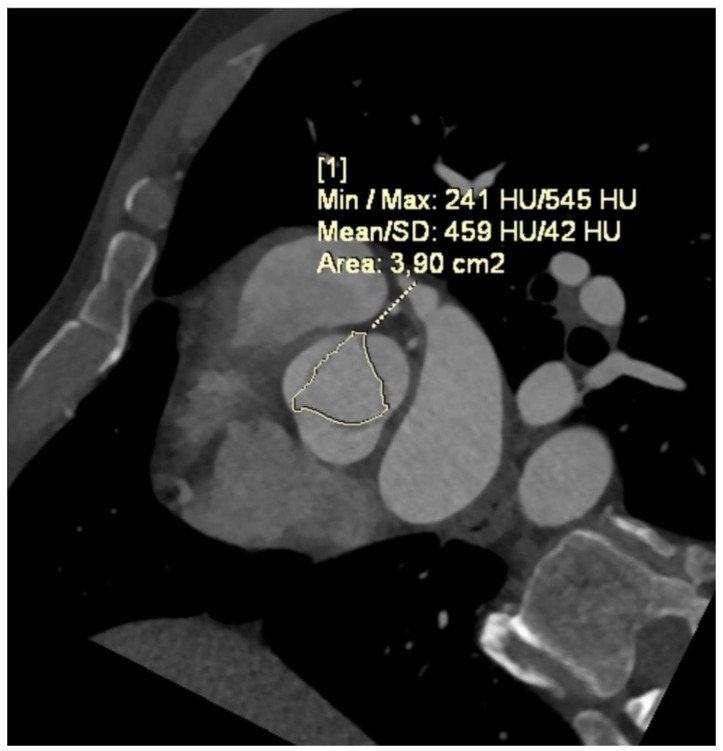
AVA measurement in an example computed tomography cardiac image (from P.G.’s clinical practice). In the application from which the image originates, [1] signifies a measurement made, which is described in the following text.

**Figure 4 jcm-13-04064-f004:**
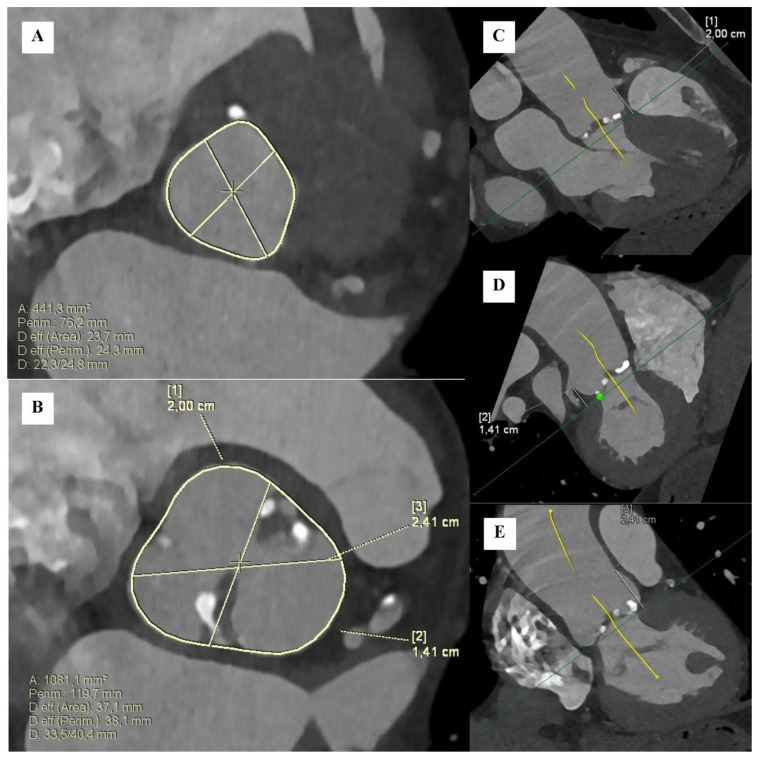
Elementary measurements in cardiac computed tomography before TAVR: (**A**) Dimensions, circumference, and surface area of the aortic annulus. (**B**) Dimensions, circumference, and surface area of the aortic bulb. (**C**) Distance of the RCA origin from the aortic annulus. (**D**) Distance of the LM origin from the aortic annulus. (**E**) Aortic bulb height (images from P.G.’s clinical practice).

**Table 1 jcm-13-04064-t001:** CT-AVCS threshold gender differences.

Authors	Male [AU]	Female [AU]
Clavel et al. (2013) [93]	2065	1274
ESC guidelines (2017) [94]	2000	1200
Pawade et al. (2018) [32]	2062	1377
Boulif et al. (2021) [95]	2238	1569
ESC guidelines (2021) [79]	3000	1600

**Table 2 jcm-13-04064-t002:** CT-AVCS and TTE comparison.

Aspect	CT-AVCS	TTE
Morphological assessment	Better assessment of leaflets, annulus, and valve anatomy; quantification of valvular calcium deposits.	Hemodynamic assessment of aortic stenosis; examination of valve function and hemodynamics.
Dependency on flow	Independent of flow.	Dependent on flow.
Accessibility	Less readily accessible and exposes patients to radiation.	Easily accessible and cheaper.
Predictive Value	Powerful predictor in the progression of aortic stenosis severity, death risk, and the future need for TAVR or TAVI.	Risk assessment and classification of aortic stenosis subtypes based on MG, AVA, PSV, and EF.
Resolution of Inconclusive Cases	Solves inconclusive cases of severe aortic stenosis with discordant echocardiography data, low-flow low-gradient (LF-LG), and paradoxical LF-LG patients.	A significant group of patients with discordant hemodynamic values can be misinterpreted.
Imaging Limitations	No imaging of fibrosis in non-contrast CT assessment; need for different cutoff values based on gender.	Hemodynamic results are independent of the etiology of aortic stenosis, for example, in rheumatic aortic disease.
Risk Stratification	New method of risk stratification in aortic stenosis; strong predictor of outcomes even in late-stage disease.	Gold standard method of risk assessment based on hemodynamics; excellent in early-stage disease but less effective in severe aortic stenosis with varying patterns of gradient and blood flow.
Quantification Limitation	Possible quantification of extravalvular calcium leading to overestimation of disease severity.	Patient comorbidities, alterations in blood flow, as well as technical faults can result in discordant echocardiographic data.
Reproducibility and Sensitivity to Changes	Higher reproducibility and greater sensitivity to minor changes over time compared to TTE.	A more operator-dependent method; possible variation in results obtained by different physicians.

## Data Availability

Not applicable.

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
