# Peer review of "Aortic Valve Calcium Score: Applications in Clinical Practice and Scientific Research—A Narrative Review"

_jcm, 2024, doi:10.3390/jcm13144064_

Round 1
Reviewer 1 Report
Comments and Suggestions for Authors
Paweł Gać et al. present a review article with the aim to investigate the essential role played by the computed tomography Aortic Valve Calcium Score (AVCS) in the cardiovascular diagnostic landscape. The article offers interesting concepts and useful advice for clinical practice. However, key inputs need to be considered to improve the quality and scientific impact of the manuscript.
1. I suggest to better discuss and show the role of multimodality imaging in the diagnosis of CAVS in the Introduction.
2. More details on the role of Lp(a) in the mechanisms of Aortic Valve Calcification should be highlighted in the paragraph 4.
3. In the paragraph “9.3. Calcium Score, TAVR and The Risk Of Stroke”, it is suggested to offer some information on diagnosis and treatment in case of stroke.
4. In the paragraph 9.5, I suggest to add and better discuss the role of aortic valve calcium scoring in predicting paravalvular aortic regurgitation after transcatheter aortic valve implantation, as already well documented in the literature. Please, include some information about diagnosing this type of aortic regurgitation.
Please, add this reference to improve these new sentences: Siani A, et al. Aortic regurgitation: A multimodality approach. J Clin Ultrasound. 2022 Oct;50(8):1041-1050. doi: 10.1002/jcu.23299.
Reviewer 2 Report
Comments and Suggestions for Authors
The authors have chosen the topic of Aortic Valve Calcium Score and its current and future potential role in diagnostics and management of aortic stenosis. They went through about a 200 articles and dealt with a number of issues, from aortic stenosis etiology to AVCS advantages and limitations, further on to various aspects of aortic stenosis assessment, etc. way too long to the role of AVCS in future scientific projects and AI implementation.
The authors definitely gathered an extensive body of information. They have chosen the genre of a narrative review which enables the authors to give their „subjective“ overview of the field and should serve in educational way to the readers. Nevertheless, as a such, it also has to keep some sound and balanced adherence to the real clinical life.
Difficult to judge from the authors affiliations but from the point of reviewer (who is active cardiac surgeon with hundreds of aortic valve surgeries performed and a member of Heart Team deciding on aortic valve disease treatment on daily basis) some basic truths have to be reminded. The only treatment of a significant aortic stenosis is surgical valve replacement or TAVI which is currently performed with very low procedural risk and little delay. The diagnosis is made straightforward easy and very, very few cases remain debatable. Principal role is the one of indexed effective orifice area (poorly mentioned in the paper!), visual echo aspect of the valve and clinical situation of the patient (NYHA etc.) Aortic stenosis at surgery may be slit-like narrow but the amount of calcium in the leaflets and the annulus and aortomitral curtain may vary considerably. For surgery, the calcium is not a problem, for TAVI the presence of calcium is a technical prerequisite to anchor the TAVI valve but TAVI complications are related to a genuine valve morfology, its cuspidity, symmetricity etc. Late results after TAVI are related to complex cardiovascular profile and comorbidities of the patient and other complications like endocarditis etc. The paper should benefit from incorporating these and other sound aspects of real clinical life, not only citing works not/documenting some correlations.
Writing: The paper is somewhat unbearably long and difficult to read. There is a plenty of unnecessary sentences repeating over and over phrases about potential roles etc. Not all the 200 works have to be commented per piece, repeatedly pro and con. It is a role of the author to extract some substantial common and prevailing features characterising the present body of knowledge.
Dividing in the parts is good, some parts are better, some are less. There should be some pronounced idea at the end of each part. Shortening into a more concise text would be appreciated.
Round 2
Reviewer 1 Report
Comments and Suggestions for Authors
The authors responded satisfactorily to my comments and doubts, congratulations
Reviewer 2 Report
Comments and Suggestions for Authors
Satisfied with the corrections.